# A Comparative Investigation of Properties of Metallic Parts Additively Manufactured through MEX and PBF-LB/M Technologies

**DOI:** 10.3390/ma16145200

**Published:** 2023-07-24

**Authors:** Janusz Kluczyński, Katarzyna Jasik, Jakub Łuszczek, Bartłomiej Sarzyński, Krzysztof Grzelak, Tomáš Dražan, Zdeněk Joska, Ireneusz Szachogłuchowicz, Paweł Płatek, Marcin Małek

**Affiliations:** 1Institute of Robots & Machine Design, Faculty of Mechanical Engineering, Military University of Technology, General Sylwester Kaliski Street 2, 00-908 Warsaw, Poland; katarzyna.jasik@student.wat.edu.pl (K.J.); jakub.luszczek@wat.edu.pl (J.Ł.); bartlomiej.sarzynski@wat.edu.pl (B.S.); krzysztof.grzelak@wat.edu.pl (K.G.); ireneusz.szachogluchowicz@wat.edu.pl (I.S.); 2Faculty of Military Technology, University of Defence, 66210 Brno, Czech Republic; tomas.drazan@unob.cz (T.D.); zdenek.joska@unob.cz (Z.J.); 3Institute of Armaments Technology, Faculty of Mechatronics, Armaments and Aerospace, Military University of Technology, General Sylwester Kaliski Street 2, 00-908 Warsaw, Poland; pawel.platek@wat.edu.pl; 4Institute of Civil Engineering, Faculty of Civil Engineering and Geodesy, Military University of Technology, General Sylwester Kaliski Street 2, 00-908 Warsaw, Poland; marcin.malek@wat.edu.pl

**Keywords:** Laser Beam Powder Bed Fusion of Metals, material extrusion, selective laser melting, Fused Deposition Modelling, Fused Filament Fabrication, 316L steel

## Abstract

In this study, the research on 316L steel manufactured additively using two commercially available techniques, Material Extrusion (MEX) and Laser Powder Bed Fusion of Metals (PBF-LB/M), were compared. The additive manufacturing (AM) process based on powder bed synthesis is of great interest in the production of metal parts. One of the most interesting alternatives to PBF-LB/M, are techniques based on material extrusion due to the significant initial cost reduction. Therefore, the paper compares these two different methods of AM technologies for metals. The investigations involved determining the density of the printed samples, assessing their surface roughness in two printing planes, examining their microstructures including determining their porosity and density, and measuring their hardness. The tests carried out make it possible to determine the durability, and quality of the obtained sample parts, as well as to assess their strength. The conducted research revealed that samples fabricated using the PBF-LB/M technology exhibited approximately 3% lower porosity compared to those produced using the MEX technology. Additionally, it was observed that the hardness of PBF-LB/M samples was more than twice as high as that of the samples manufactured using the MEX technology.

## 1. Introduction

Additive Manufacturing is an increasingly utilized technology for producing a wide diversity of components [1]. Although initially used solely for prototyping purposes, with polymers being the most commonly used materials, it is now increasingly employed for manufacturing functional parts applied in demanding branches of industry such as aviation, automotive, energy, bioengineering, and medicine [2,3,4,5]. The expansion of additive manufacturing technologies is associated with the advantages it offers in comparison to conventional manufacturing methods [6]. The key advantages include the possibility of fabrication of complex geometric elements, design freedom, material waste minimization, and the elimination of tools like molds or dyes used in a conventional technology of production. Furthermore, progress in AM techniques resulted in the development of machines wherein parts are made from different types of metals or even composite materials. They are used for the production of “ready-to-use” parts with improved final physical and mechanical properties [7]. Due to its favorable characteristics, such as high strength, high ductility, corrosion resistance, and biocompatibility, stainless steel 316L is one of the most popular materials used in metal additive manufacturing (MAM) techniques [8,9,10,11,12].

Currently, one of the most popular groups of metal additive techniques dedicated to manufacturing metal components is (Laser Based Metal Powder Bed Fusion) PBF-LB/M, where Selective Laser Melting (SLM) is one of the most popular solutions within this technique. It involves the deposition of metallic powder layers and their fusion using a high-energy source like laser light focused on a small surface of melted material [13,14,15,16]. The properties of parts produced using this method are comparable to conventionally manufactured parts [17]. To date, there has been a lot of research on SLM-printed 316L steel. Kong et al. [18] investigated the effects of process parameters on microstructure and mechanical properties, as well as the corrosion SLM 316L. Sun et al. [19] improved the scanning speed to produce SLM SS 316L alloy with the highest possible density at low porosity. Bartolomeu et al. [20] compared the mechanical properties of 3D-printed 316L stainless steel using three different technologies: SLM, hot pressing, and conventional casting. The results showed that the best mechanical properties, such as hardness and tensile strength, were achieved for samples produced using the SLM technique. In studies conducted by other researchers [21,22], the influence of SLM process parameters on the microstructure, mechanical properties, corrosion resistance, and biocompatibility of 316L steel was examined. Based on the presented results it was possible to state that samples manufactured with the use of higher values of the laser power exhibited improved strength properties, greater corrosion resistance, and enhanced biocompatibility. The properties of metal parts produced by AM techniques can be improved by applying additional heat treatment such as Hot Pressing Isostatic HIP. Cegan et al. [23] in their study showed that HIP caused a significant decrease in the internal closed porosity of SLM-manufactured austenitic steel 316L samples to 0.1%, Samples after HIP showed lower yield strength than after SLM (from 290 to 325 MPa) and relatively high ductility of 47.8–48.5%, regardless of the SLM conditions used. Despite many advantages of the SLM AM technique, there is one significant drawback related to high equipment purchase and operational costs. Therefore, more cost-effective metal additive manufacturing methods are being introduced, such as the MEX technique, with the most common methods being Fused Deposition Modeling/Fused Filament Fabrication (FDM/FFF) [24,25,26,27]. While this method of production is most commonly used for polymer materials, 3D printing processes’ advantages have led to its increasing utilization where materials like metal powders have been started to be used. Implementation of metal-based materials to the FFF technique required special composite filaments that consist of metallic powder and polymer matrix. However, to obtain parts consisting of pure metal, the 3D-printed parts undergo a process called catalytic debinding, followed by sintering to eliminate the binding phase. BASF 316L is an example of such a material. In their works, authors [28,29] examined the influence of printing direction on the performance properties of elements printed using BASF 316L material. Static tensile tests showed that the samples exhibited a similar failure process, except for tensile strength and elongation at break. Decker et al. [30] investigated the strength properties of 316L steel printed using the FDM/FFF technology and compared the results with those of 316L steel printed using the SLM technology. A significant decrease in tensile strength and fatigue strength was observed for samples printed with FDM/FFF compared to the SLM technology. Quarto et al. [31] investigated selected printing parameters to improve the performance properties of printed parts, minimize their porosity, and examine dimensional shrinkage.

Due to the limited availability of studies describing the 3D printing process of metals, including 316L steel, using the FDM/FFF technology, further research, process phenomena description, and potential improvements in the material’s performance properties are needed. That is why this paper aims to determine if FDM/FFF techniques could state an alternative to much more expensive AM techniques like SLM, and the main pros and cons of both techniques. The paper compares properties such as microstructure, hardness, and roughness of two metal printed parts using MEX and PBF-LB/M technologies to point out the potential areas of application of both AM technologies.

## 2. Materials and Methods

### 2.1. Materials

Carpenter Additive Company’s (Carpenter Additive, Widnes, UK) 316L stainless steel powder was used for sample production via the SLM technology. The powder particles exhibited a spherical shape with a diameter ranging from 15 to 63 μm. The chemical composition of the powder is presented in Table 1.

The second material used for the production of material specimens with the use of the FDM/FFF 3D printing technique was BASF Ultrafuse 316L filament (BASF, Ludwigshafen am Rhein, Germany). This material is in the form of a polymer composite combined with 316L stainless steel powder. Ultrafuse 316L contains 90% stainless steel in its composition. The material can be utilized on a standard FDM/FFF printer. Prints from this material require additional post-processing. Firstly, the printed parts need to undergo a debinding process to remove the polymer binder. This is carried out in specialized equipment through a thermochemical catalytic process, in which the prints are exposed to nitrogen oxide fumes. After debinding, the prints are sintered at a high temperature of approximately 1400 °C.

### 2.2. AM Processes Description

The 3D models of the test samples were designed using SolidWorks (Dassault Systems; Waltham, QC, Canada) CAD software (version 2022–2023). The first type of SLM samples was produced on an SLM 125HL machine (SLM Solutions AG, Lübeck, Germany). The samples had a cubic shape, as it is shown in Figure 1. The samples were rotated by an angle of 30° to prevent damage to the support structures in the additive manufacturing device. For the sample’s production process via SLM, the default setting was used: layer thickness was set at 0.03 mm, with a hatch distance of 0.12 mm. The power level was set to 200 W, and the scanning speed was 800 mm/s. Based on these parameter values, the energy density for the exposure was calculated, resulting in a value of 69.4 J/mm^3^. The calculation was performed using the following formula:$$\mathrm{VED}=\frac{P_{L}}{V_{s}{*h}_{d}{*L}_{T}}$$

VED—Volumetric Energy Density (J/mm^3^),P_L_—laser power (W),V_s_—scanning speed (mm/s),h_d_—hatching distance (mm),L_T_—layer thickness (mm).

In the next step, the print preparation was carried out, the 3D model was saved in a stereolithography tessellated file (.stl). A job file was prepared using the Magics software (v19, Materialise, Leuven, Belgium). The finalized files were imported into the additive manufacturing device. The platform height of the device was adjusted, and the first layer of powder, known as the “zero level”, was spread. The laser wavelength used during the manufacturing process was 1080 nm.

**Figure 1 materials-16-05200-f001:**
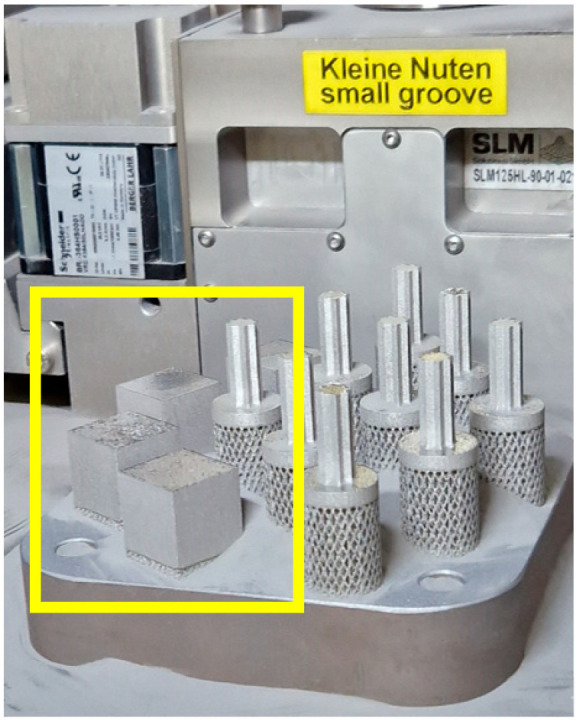
The samples (indicated by the yellow outline) after completion of the printing process.

The subsequent 3D printing method used in these studies was FDM/FFF with additional thermos-chemical treatment (shown in Figure 2). It was applied to produce cuboid-shaped material specimens with dimensions presented in Figure 4b. Operation codes (g-codes) for the printing process were prepared in the dedicated Prusa Slicer v2.5.2 software. The 3D printing process was carried out using a Prusa i3 MK3s 3D (Prusa Research, Prague, Czech Republic) printer. The process parameters (shown below) were adjusted according to the material’s producer recommendations:Filament diameter: 1.75 mm,Nozzle diameter: 0.4 mm,Nozzle temperature: 250 °C,Bed temperature: 100 °C,Infill: 100%,Number of contours: 5.

After the 3D printing process, the material samples were subjected to additional post-processing to obtain the required mechanical and physical properties. This process consists of catalytic debinding and sintering stages, which were performed directly by the material manufacturer as an external service.

**Figure 2 materials-16-05200-f002:**
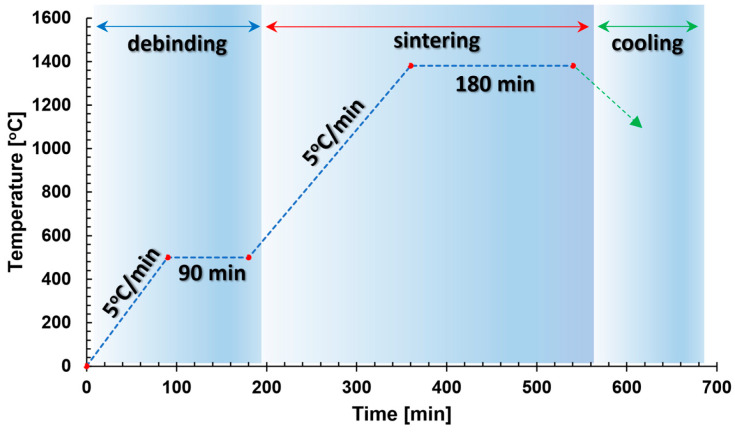
Thermal cycle of debinding and sintering process of 316L BASF Ultrafuse material.

Material samples after the manufacturing process were subjected to microstructural analysis to evaluate the microstructure quality as well as verify the presence of material imperfections like porosity, voids, or cracks. For this purpose, samples were cut parallel to the printed layers, embedded, and then ground with sandpaper of grades 320, 500, 800, 1200, and 2400. Subsequently, the samples were polished using a neoprene cloth with the addition of water and OP-S solution. In the final stage, the samples were subjected to etching. Etching was performed in a digestion unit using an acetic glycerol solution (6 mL HCl, 4 mL HNO_3_, 4 mL CH_3_COOH, and 0.2 mL glycerol) as the etchant. First, the porosity of the details was examined by measuring the density according to Archimedes’ (Figure 3) principle:$$p=\frac{\rho_{c}-\rho_{t}}{\rho_{c}}*100\%$$

p—porosity [%],ρ_c_—density of conventional material [g/cm^3^],ρ_t_—density of test material [g/cm^3^].

**Figure 3 materials-16-05200-f003:**
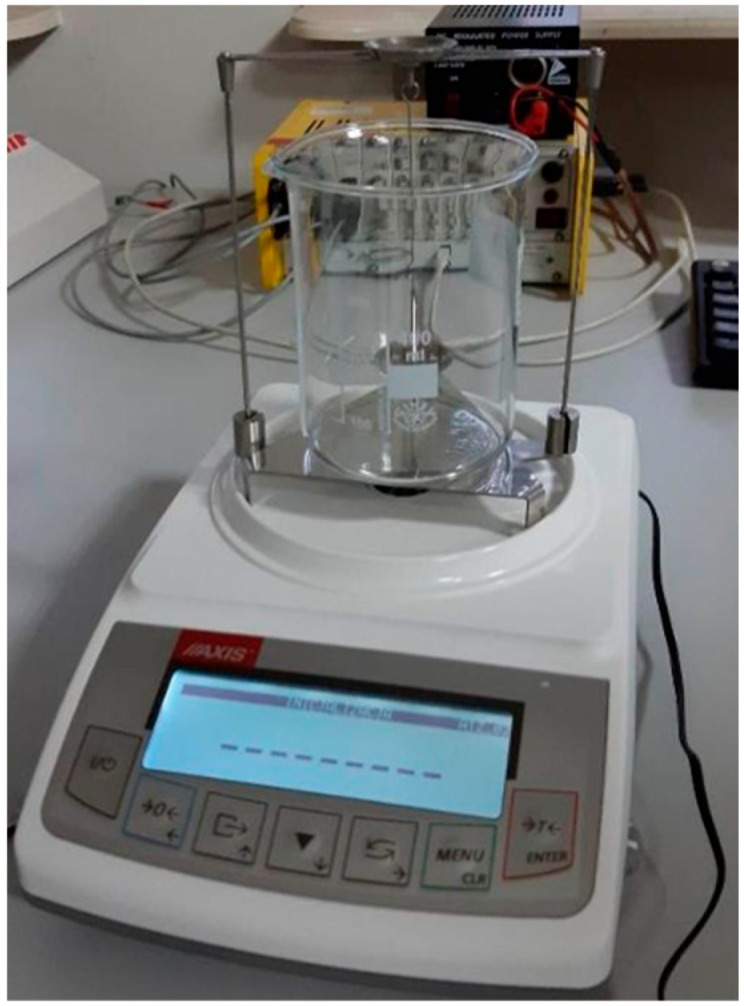
The measurement setup for the density measurement of the samples is shown in Figure 1.

The study was conducted for three types of 316L stainless steel samples—Conventionally Manufactured (CM), using SLM, and FDM/FFF 3D printing techniques. The dimensions of each sample are shown in Figure 4. During the research five samples of each type were analyzed to check the result’s repeatability. From the group of three samples with repeatable results, it has been selected results from an exact sample.

The microstructure analysis and additional porosity measurements of the samples were conducted using an Olympus 4100 LEXT (Shinjuku, Tokio, Japan)) confocal microscope (Figure 5) with consideration of four types of material samples. The samples build additively with the use of SLM and FDM/FFF technologies were cut in two planes—along the direction of layer deposition during the 3D printing process (plane 0XY in Figure 4) and along the fabricated layers (plane 0YZ in Figure 4).

Subsequently, to verify the accuracy of the measurement results obtained using the Archimedes method, the porosity in individual sample sections was measured using dedicated software (Mountains Map, version 7.0). The porosity was calculated based on the average number of grains for each sample. Then, microhardness measurements were performed using the Vickers method according to the PN-EN ISO 6507-1 standard [32], using a Struers DuraScan 70 microhardness tester (Ballerup, Denmark) (Figure 6).

The measurements were carried out using a diamond indenter with a regular tetrahedral shape and an apex angle of 120°. Six measurements were conducted for each sample, and two extreme results were discarded for subsequent calculations. The distance between individual measurements was three times greater than the diameter of the indentation to ensure that the results did not influence each other. At the final stage, surface roughness measurements were made by means of the Keyence VHX7000 digital microscope (Osaka, Japan) (Figure 7).

## 3. Results and Discussion

### 3.1. Density

Density measurements were performed for three different types of material samples, and the gathered results are presented in Table 2.

Based on the obtained results, it was observed that samples of conventionally manufactured 316L steel exhibited the highest density and the lowest porosity. The density value was determined to be 7.94 ± 0.01 g/cm^3^. On the other hand, samples produced using the FDM technology showed the lowest density, with a value of 7.67 ± 0.02 g/cm^3^, which was 3.37 ± 0.24% lower than the value obtained for conventionally manufactured steel. Meanwhile, the density of the samples produced using the SLM technology was determined to be 7.90 ± 0.02 g/cm^3^, which was 0.48 ± 0.24% lower than that of conventionally manufactured samples.

### 3.2. Porosity

To determine the nature of the recorded porosity, further analyses were conducted using a confocal microscope. Five measurements were performed for each sample, and one image for each sample is presented below (Figure 8).

In Figure 8a,c, clusters of porosity associated with gaps between the deposited material paths can be observed in the FDM/FFF sample. In the case of SLM samples, it can be noticed that the porosity is significantly lower, with pores having smaller volumes and occurring throughout the structure of the part in a stochastic manner. Analyzing the images taken for samples cut through the layers along the 0Z-axis, it is also evident that the FDM sample exhibits significantly higher porosity compared to the SLM counterpart. For both types of samples, the pore size is irregular, but it is considerably smaller for SLM samples than for FDM/FFF samples. In the case of FDM/FFF technology, these defects can be reduced by adjusting 3D print parameters such as scanning speed and print temperature, or by employing post-processing methods such as hot isostatic pressing (HIP) [33]. Based on the captured images, the porosity was calculated based on the average grain count for each sample. The results are presented in Table 3.

The FDM/FFF-printed samples exhibited higher porosity compared to the SLM-printed samples. The highest porosity value of 3.65% was observed for the FDM/FFF sample analyzed in the layer-by-layer direction (0XY plane). The porosity for FDM/FFF samples, measured along the printed layers, was 3.13%, which was 14.2% lower than the porosity measured in the 0XY plane. What is more, there is a significant value of the standard deviation of both tested surfaces in FDM/FFF samples, which is related to a local defect caused by a nonproper connection between extruded material in green parts (before debinding and sintering). On the other hand, the porosity values for SLM samples were significantly lower, with 0.40% for the 0YZ plane and 0.13% for the 0XY plane. These results align with the porosity values obtained through Archimedes’ principle measurements. The higher porosity of FDM/FFF printed samples is due to the way the material is applied. Layers of material are fused and applied on top of each other. Adhesion between successive layers can be hindered due to differences in temperature and surface properties between each layer. This can lead to a weaker bond between layers and the formation of porous areas. The study [34] proves that samples produced by the FDM/FFF technique also had higher porosity than counterparts produced by SLM. The porosity value was higher by about 1.5%.

### 3.3. Microstructural Investigation

Based on the analysis of the images of the material specimens manufactured with the use of the FDM/FFF technique, it is evident that empty spaces form between the printed outline shells, as visible in Figure 9. The outer outlines diverge, creating non-connected structures at the edges. However, the layers stacked upon each other in the 0Z direction form a cohesive structure, as shown in Figure 10. The material structure exhibits a relatively regular morphology, with individual material particles having similar shapes and sizes.

The microstructure images of the SLM samples reveal the distinctive melt tracks characteristic of the SLM process, which are oriented at specific angles for each layer. The presence of voids within the structure is also visible. The irregular shape of these voids may indicate a local lack of fusion, which can be associated with porosity that exists in the case where a lack of fusion (LOF) is observed (Figure 11). These observations suggest that the SLM process introduces specific microstructural features, including the aligned melt tracks and the potential occurrence of LOF-related porosity.

This defect is a structural flaw primarily caused by a local lack of sufficient input energy during the melting process. The formation of Lack of Fusion (LOF) is attributed to the fact that metal powders are not fully melted to deposit a new layer onto the previous one with sufficient bonding [35]. In the area where the defect occurs, the surface becomes more rough, leading to a change in the wetting angle of the surface. This hinders the smooth flow of the molten pool, resulting in such interlayer defects.

### 3.4. Hardness Analysis

Hardness testing was conducted on SLM and FDM/FFF samples in two planes. The applied load during the HV0.5 testing was 4.9 N. The results are presented in Table 4.

The SLM samples exhibited higher hardness. The value was the same for both planes and amounted to 246.8 HV0.5. The FDM/FFF samples had over two times lower hardness. For the XY plane, the value was 120.3 HV0.5, while for the 0YZ plane, it was slightly lower at 118.4 HV0.5. The significant drop in hardness for the FDM/FFF samples can be attributed to additional heat treatment as part of the Ultrafuse 316L material fabrication process. No additional post-processing treatments were performed on the SLM samples made from 316L steel, which explains their high hardness values [36]. FDM/FFF printed samples are characterized by bigger porosity than SLM printed samples, which have a more homogeneous structure with significantly smaller amounts of pores.

### 3.5. Surface Roughness Analysis

The research results with measured main roughness parameters are shown in Table 5. The measurements for all specimens were conducted in the same manner. The total measured length of the profiles is indicated by blue points (X–signs), the length taken into account for roughness calculation is indicated by red color in Table 5. In the case of the 0XY plane, the average surface roughness (R_a_) is at the same level in both analyzed samples’ groups. In the case of 0YZ, the FDM/FFF samples indicated an R_a_ parameter almost twice lower. Based on the typical procedures for parts obtained by metal AM the samples should be subjected to additional surface treatment by sandblasting, which would lead to obtaining the same levels of surface condition regardless of the printing direction [37]. In the study [31], the surface roughness R_a_ of the FFF 316L metal samples was 7.5 μm is much higher than that of the SLM samples, which was 5.8 μm. However, the differences in surface roughness results are influenced by the printing parameters used, the printing strategy, as well as the type of fiber used.

## 4. Conclusions

Based on the conducted research it was possible to identify the potential microstructural and mechanical differences between three types of 316L stainless steel obtained in a different manufacturing process, a typical metallurgical process, and two 3D printing processes like SLM and FFF/FDM. Obtained results allow the conclusion that 316L steel material samples build additively using the SLM technology exhibit superior microstructural properties compared to those produced using the FDM/FFF technology. The most important difference was registered in the material microstructure, where the SLMed parts in an as-built condition were constituted on solidified molten pools, and FDM/FFF-ed samples were made on bonded material particles. Each microstructure was affected by some characteristic features that are strictly related to each of the considered AM techniques. The conducted research allowed for the following outcomes to be drawn:
(1)The SLM-ed samples have significantly lower porosity, with a difference of over 3% compared to the FDM/FFF samples. This result is consistent with both porosity measurement methods employed in this study.(2)The microstructure of the SLM samples appears more solid and dense compared to the FDM/FFF samples, indicating a higher degree of material consolidation.(3)The hardness of the SLM samples is more than twice as high as that of the FDM/FFF samples. This indicates that the SLM-printed 316L steel possesses greater hardness and potentially better mechanical properties.(4)The surface condition of samples obtained via each AM technology is strictly related to process characteristics. There are visible typical artifacts of each method (extrusion paths in FDM/FFF samples and scanning lines in SLM samples). Despite differences between both AM technologies, the surface roughness was almost at the same condition in the case of R_a_ parameters measured on individual surfaces.

The most crucial findings of the conducted research suggest that the SLM technology yields superior structural characteristics, including lower porosity, denser microstructure, and higher hardness, in comparison to the samples obtained via FDM/FFF. At the same time, it is visible that there is a significant field to use cheaper and more accessible FDM/FFF technology after introducing additional postprocessing. Such an approach would ensure a better quality of the obtained parts, and make the FDM/FFF metallic parts more competitive with their SLM-made counterparts.

## Figures and Tables

**Figure 4 materials-16-05200-f004:**
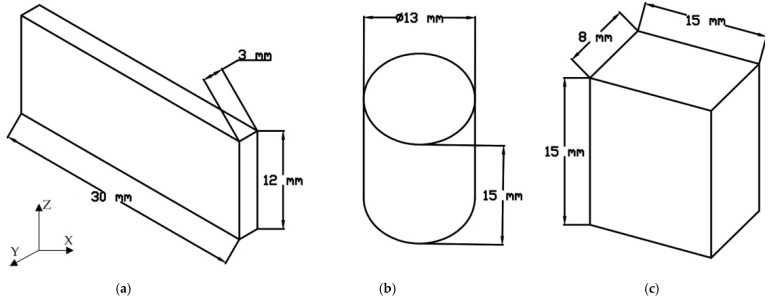
The samples used for density measurement: (**a**) Conventionally manufactured (CM); (**b**) FDM/FFF; (**c**) SLM (Selective Laser Melting).

**Figure 5 materials-16-05200-f005:**
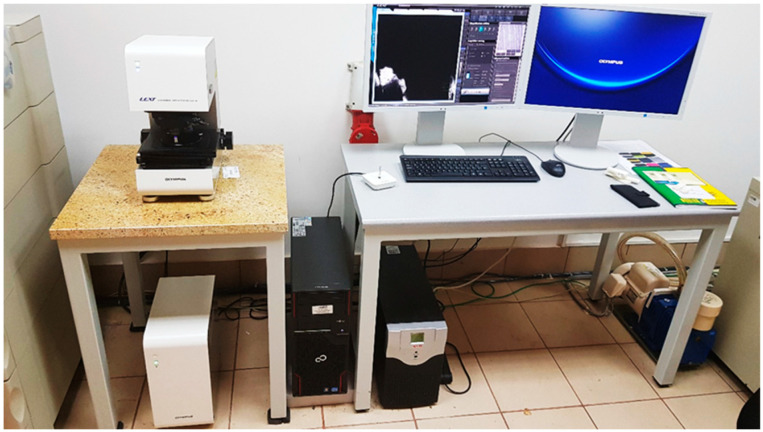
Olympus 4100 LEXT confocal microscope used in the research.

**Figure 6 materials-16-05200-f006:**
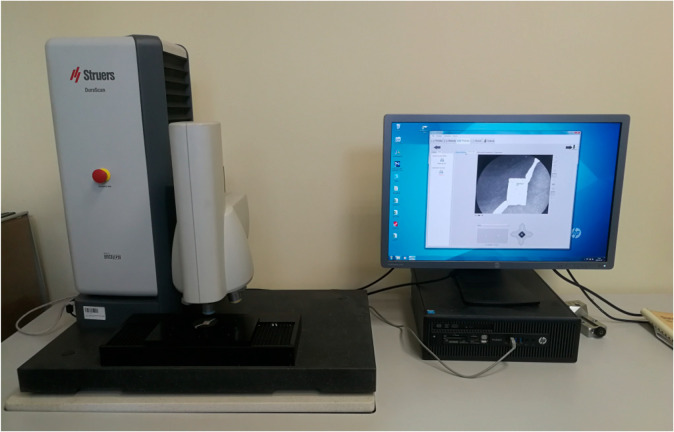
Struers DuraScan 70 hardness tester used in the research.

**Figure 7 materials-16-05200-f007:**
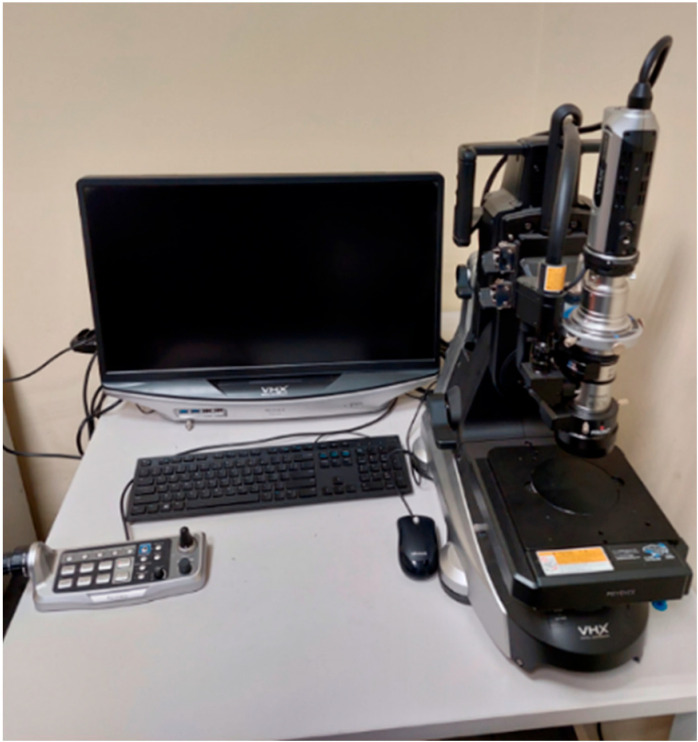
Keyence VHX7000 digital microscope used in the research.

**Figure 8 materials-16-05200-f008:**
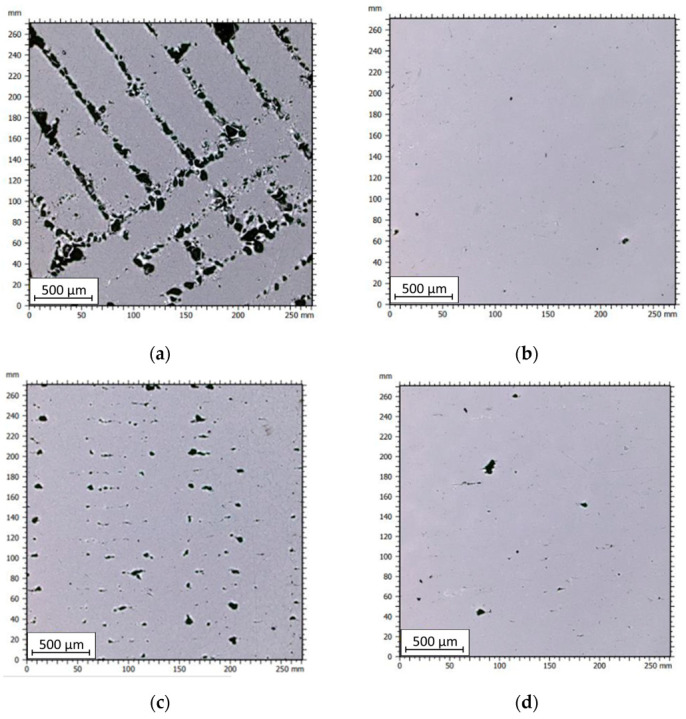
Images of the samples captured under the microscope: (**a**) FDM sample (0XY plane); (**b**) SLM sample (0XY plane); (**c**) FDM sample (0YZ plane); (**d**) SLM sample (0YZ plane).

**Figure 9 materials-16-05200-f009:**
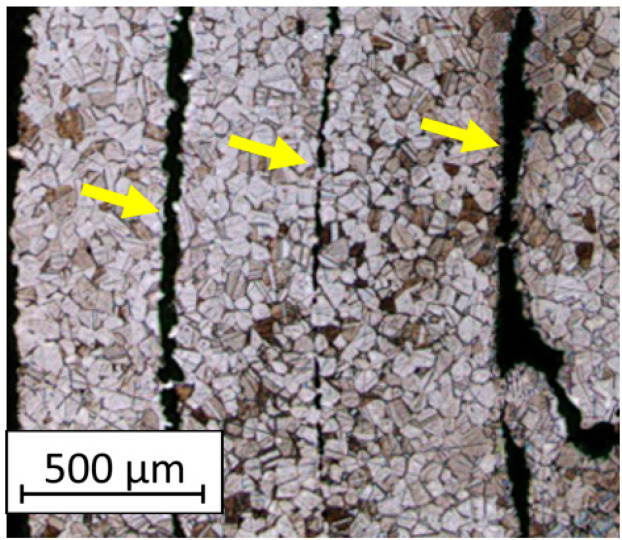
The microstructure of the FDM/FFF samples (cross-section taken in the 0XY plane) with marked unfused material paths (yellow arrows).

**Figure 10 materials-16-05200-f010:**
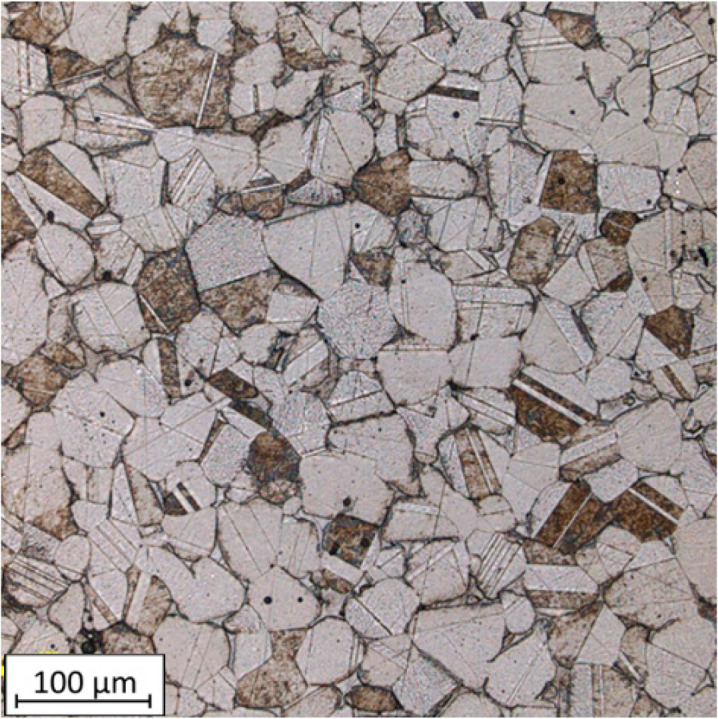
The microstructure of the FDM/FFF samples (cross-section taken in the 0YZ plane).

**Figure 11 materials-16-05200-f011:**
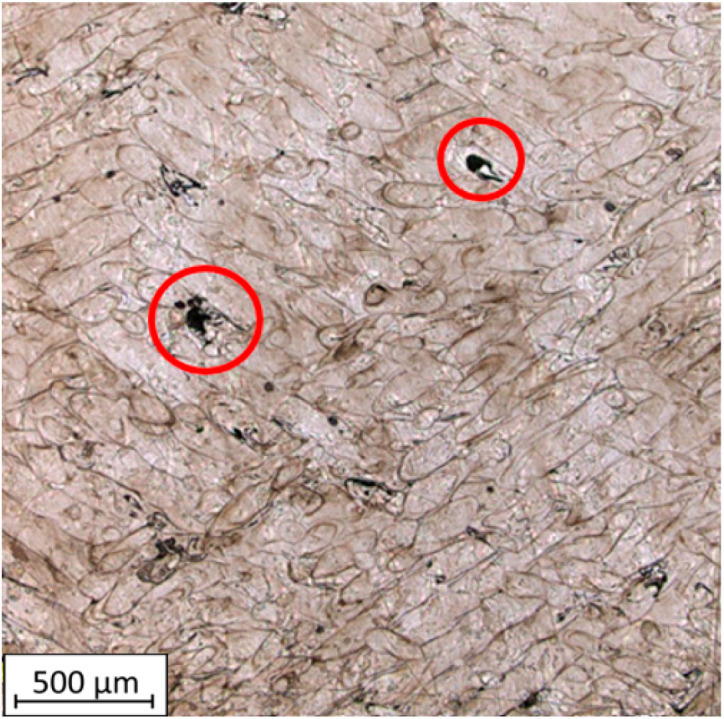
Microstructure of SLM samples with highlighted LOF voids (red color circles).

**Table 1 materials-16-05200-t001:** Chemical composition of 316L steel metal powder distributed by Carpenter Additive.

C	Cu	Mn	Si	O	P	S	N	Kr	Pn	Ni
Weight (%)										
0.027	0.02	0.98	0.72	0.02	0.011	0.004	0.09	17.8	2.31	12.8

**Table 2 materials-16-05200-t002:** Density and porosity results of the samples.

Type of Test Sample	Density (g/cm^3^)	Porosity (%)
CM	7.94 ± 0.01	~0
SLM	7.90 ± 0.01	0.48 ± 0.28
FDM/FFF	7.67 ± 0.02	3.37 ± 0.24

**Table 3 materials-16-05200-t003:** Pores volume measurement results of all tested samples.

Sample	The Surface Area of the Test Sample	Pores Area (mm^2^)	Pores Area (%)
FDM/FFF (XY)	73,846	2676 ± 1630	3.65 ± 2.22
SLM (XY)	73,315	96 ± 32	0.13 ± 0.04
FDM/FFF (YZ)	73,354	2296 ± 962	3.13 ± 1.30
SLM (YZ)	73,500	294 ± 116	0.40 ± 0.16

**Table 4 materials-16-05200-t004:** Hardness measurements.

Type of Test Sample	HV0.5 (XY)	HV0.5 (YZ)
SLM	247.33 ± 5.96	246.83 ± 2.79
FDM/FFF	129.50 ± 10.11	130.00 ± 9.00

**Table 5 materials-16-05200-t005:** Surface roughness measurements.

Sample’s Type	SLM—0XY Plane	SLM—0YZ Plane	FDM/FFF—0XY Plane	FDM/FFF—0YZ Plane
Surface image with the indicated profile line	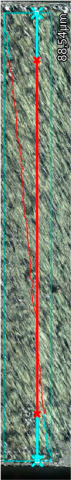	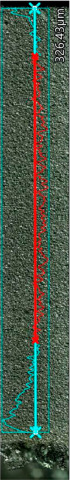	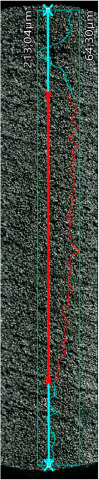	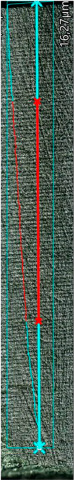
Measured profile length[μm]	18,960.35	18,745.00	11,881.16	18,468.25
Measured profile height[μm]	722.40	93.44	148.74	849.73
R_z_[μm]	25.70	31.72	35.59	28.40
R_a_[μm]	5.66	7.02	5.61	3.90

## Data Availability

The data could be available on demand.

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
