# Peer review of "A Comparative Investigation of Properties of Metallic Parts Additively Manufactured through MEX and PBF-LB/M Technologies"

_materials, 2023, doi:10.3390/ma16145200_

Round 1
Reviewer 1 Report
The paper is interesting but must be improved in some parts.In particular:
-The purpose of the research must be clearly expressed in the introduction. In the present form, it is not sufficiently described.
-I suggest using commonly used terms in literature like Hatch Distance, Scanning Speed etc, instead of Vectors distance, and printing Speed.
-Line 107-109-The choice of energy range for printing parameters should be more justified. Are there any previous studies that indicate the goodness or reasoning of the authors?
-I suggest specifying the software used for printing preparation
-Microstructure investigation – lack of fusion should be better related to literature (is it spherical porosity?)
-Section 3.5. Surface roughness analysis and Table 5 are repeated twice
-The last part of the document: conclusions and references should follow the formatting of the journal
none
Author Response
Dear Reviewer,
On behalf of all authors, I would like to thank you for taking the time to read our manuscript and put your comments which allowed us to improve the quality of our work. Please note that all corrections made thanks to your comments were green–highlighted. Below you can find our answers related to each of your comments.
- The purpose of the research must be clearly expressed in the introduction. In the present form, it is not sufficiently described.
Ad.1. We have added a clearer research purpose to the last part of the introduction.
- I suggest using commonly used terms in literature like Hatch Distance, Scanning Speed etc, instead of Vectors distance, and printing Speed.
Ad.2. We corrected the mentioned issues.
- Line 107-109-The choice of energy range for printing parameters should be more justified. Are there any previous studies that indicate the goodness or reasoning of the authors?
Ad.3. During the SLM process we used a default settings provided by the SLM Solution company – we put a proper explanation in the manuscript.
- I suggest specifying the software used for printing preparation.
Ad.4. We have added the types of software we used in preparing the prints for each method.
- Microstructure investigation – lack of fusion should be better related to literature (is it spherical porosity?).
Ad.5. We referred ,,lack of fusion" to literature and put proper citations.
- Section 3.5. Surface roughness analysis and Table 5 are repeated twice.
Ad.6. Thank you for pointing this issue out. We removed the table.
- The last part of the document: conclusions and references should follow the formatting of the journal.
Ad.7. We have improved the formatting of conclusion and references in accordance with the formatting of the journal.
Reviewer 2 Report
Pls find the comments in the attachment

Language can be improved by correcting the grammar mistakes
Author Response
Dear Reviewer,
On behalf of all authors, I would like to thank you for taking the time to read our manuscript and put your comments which allowed us to improve the quality of our work. Please note that all corrections made thanks to your comments were blue–highlighted. Below you can find our answers related to each of your comments.
- Abstract need to be rewritten neatly up to 300 words discussing brief theme of the research work conducted.
Ad.1. Based on the instruction to authors in the Materials journal, the abstract should have up to 200 words. We have rewritten our abstract according to your comments. Now it has 187 words.
- The title of the article needs to be modified as A comparative Investigation of properties of metallic parts additively manufactured through MEX and PBF-LB/M technologies.
Ad.2. Thank you very much for your advice. We have changed the title of the work on the suggested one.
- At the end of the introduction, the section novelty of the research work needs to be described in one paragraph at least. At the end of the introduction section line no.84, performance properties …. Sentence seems to be incomplete.
Ad. 3. At the end of the introduction, we added a separate summary paragraph. It was green highlighted, because the other reviewer also suggest such improvement.
- In the Materials and Methods section, the details of the test conducted for how many samples should be mentioned with proper justification.
Ad.4. We put an additional comment in the manuscript: During the research five samples of each type were analyzed to check the results repeatability. From the group of three samples with repeatable results it has been selected a results from an exact sample.
- The process parameters adjusted for the selected material were as follows:
Filament diameter: 1.75 mm,
Nozzle diameter: 0.4 mm,
Nozzle temperature: 250°C,
Bed temperature: 100°C,
Infill: 100%,
Number of contours: 5……
How these values are selected/decided??
Ad.5. The parameters were selected based on the material’s producer recommendations. We put proper information in the manuscript: “The process parameters (shown below) were adjusted according to the material’s producer recommendations”
- Result and discussion section needs to be described by giving proper citations (studies conducted by earlier researchers). Authors are suggested to refer 5 latest articles related to metal extrusion and PBF-LB/M.
Ad.6. We extended the discussion and supported it with proper citations.
- Why only porosity, hardness, surface roughness tests were conducted in this research work? How it is important from all other properties of metallic parts?
Ad.7. Such an approach was used to allow proper comparison and keep it easy to repeat– all presented research is possible to made with a digital microscope, hardness tester, and advanced weight (Archimedes density). We did not included further analysis in the case of static (tensile, compressive, bending) properties, fatigue and dynamic testing because it would require additional three combinations of orientation (to keep the reliability of the research). Such an approach would make this paper much bigger. What is more, we wanted to focus mostly on structural phenomena.
- In conclusion section, Comparative investigation about metal extrusion and PBF-LB/M needs to be described.
Ad.8. We put an additional part based on your comment and connect it with the main conclusions:
Based on the conducted research it was possible to identify the potential microstructural and mechanical differences between three types of 316L stainless steel obtained in different manufacturing processes, a typical metallurgical process and two 3D printing processes like SLM and FFF/FDM. Obtained results allows to drawn that 316L steel mate-rial samples build additively using the SLM technology exhibit superior microstructural properties compared to those produced using the FDM/FFF technology. The most important difference was registered in the material microstructure, where the SLMed parts in an as-built condition were constituted on solidified molten pools, and FDM/FFF-ed sam-ples were made on bonded material particles. Each of microstructure was affected by some characteristic features that are strictly related to each of considered AM techniques. Conducted research allowed to drawn the following outcomes:
(1) The SLM-ed samples have significantly lower porosity, with a difference of over 3% compared to the FDM/FFF samples. This result is consistent with both porosity meas-urement methods employed in this study.
(2) The microstructure of the SLM samples appears more solid and dense compared to the FDM/FFF samples, indicating a higher degree of material consolidation.
(3) The hardness of the SLM samples is more than twice as high as that of the FDM/FFF samples. This indicates that the SLM-printed 316L steel possesses greater hardness and potentially better mechanical properties.
(4) Surface condition of samples obtained via each AM technology is strictly related to process characteristics. There are visible typical artifacts of each method (extrusion paths in FDM/FFF samples and scanning lines in SLM samples). Despite differences between both AM technologies, the surface roughness was almost at the same condi-tion in the case of Ra parameters measured on individual surfaces.
The most crucial findings of the conducted research suggest that the SLM technology yields superior structural characteristics, including lower porosity, denser microstructure, and higher hardness, in comparison to the samples obtained via FDM/FFF. At the same time, it is visible that there is a significant field to use cheaper and more accessible FDM/FFF technology after introducing additional postprocessing. Such kind of approach would assure betted quality of the obtained parts, and make the FDM/FFF metallic parts more competitive to the SLM-made counterparts. Grammar check throughout the article is suggested for the authors.
Ad.9. The whole paper has been proofread once again after all the corrections. All corrections were made with the track changes tool.
- Authors have nicely conducted the comparative investigation, but unfortunately failed to correlate their results obtained with the earlier studies. So it is strongly recommended to the authors to cite at least 5-8 research articles relevant with the existing research work. Flow of article seems to be good, but the conclusion section needs to be rewritten neatly.
Ad.10. As we mentioned at point 6 we extended the discussion of the obtained results and covered it with additional citations. Also as we mentioned at point 8 – we rephrased and extended a conclusion part.
Reviewer 3 Report
The work compares the properties of metallic parts additively manufactured via MEX and PBF-LB/M. It needs the following revisions to be implemented.
The scope, novelty and objective of work is not very much clear. It must demonstrate novelty compared to existing literature.
What is the meaning of Line 83 and 84 "further research, process refinement, and improvements in the material's performance properties (were carried out ??)"
The SEM and EDS analysis of the powder should be included to confirm the size and elemental composition.
The photos of experimental setup should be included.
Figure 4, 5, 6, 7 must contain labels for proper understanding.
A technical justification about improvement in density, porosity, hardness and surface roughness must be included and compared with available literature.
Author Response
On behalf of all authors I would like to thank you for taking your time to read our manuscript and put your comments which allowed to improve the quality of our work. Please note that all corrections made thanks to your comments were yellow – highlighted. Below you can find our answers related to each of your comments.
- The scope, novelty and objective of work is not very much clear. It must demonstrate novelty compared to existing literature.
Ad.1. We have supplemented the introduction with an additional paragraph describing these issues.
- What is the meaning of Line 83 and 84 "further research, process refinement, and improvements in the material's performance properties (were carried out ??)"
Ad.2. We have corrected this sentence.
- The SEM and EDS analysis of the powder should be included to confirm the size and elemental composition.
Ad.3. We tried to make that kind of analysis but, a share of polymeric matrix in the BASF filament enabled to make such measurements. We were successful only on the powder dedicated for SLM, but we decided not to include it because it would be not possible to do any comparison to FDM/FFF.
- The photos of experimental setup should be included.
Ad.4. According your comments we put all the photos of used experimental setup (Figures 3,5,6,7)
- Figure 4, 5, 6, 7 must contain labels for proper understanding.
Ad.5. All additional labels that allow proper identification were yellow-highlighted.
- A technical justification about improvement in density, porosity, hardness and surface roughness must be included and compared with available literature.
Ad.6. We extended the discussion and supported it with proper citations.
Reviewer 4 Report
Reviewer’s comments
A comparison of the properties of metallic parts additively manufactured via MEX and PBF-LB/M technologies. The problem with this paper is results are very few. This paper can be accepted after major revision.
1) It would be nice if the authors could do more work like EBSD, Texture, TEM, and mechanical testing in this work.
2) The introduction of this paper is too short. It is also mandatory to add the following references to the text, especially for additive manufacturing of different categories of steel.
a) Fatigue strength of additively manufactured 316L austenitic stainless steel.
b) Tensile and fatigue properties of the binder jet printed and hot isostatically pressed 316L austenitic stainless steel.
c) High temperature study of the evolution of the tribolayer in additively manufactured AISI 316L steel.
d) Tensile and impact behaviour of thermo mechanically treated and micro-alloyed medium carbon steel bar.
e) Origin of dislocation structures in an additively manufactured austenitic stainless steel 316L.
3) “Due to the limited availability of studies describing the 3D printing process of metals, including 316L steel, using the FDM/FFF technology, further research, process refinement, and improvements in the material's performance properties (were carried out ??)”
Remove the word (were carried out ??).
4) The full name of CM is missing in the text.
5) The authors have not discussed why the FDM sample exhibits a higher porosity fraction.
6) Why is the hardness of FDM/FFF lower than SLM samples?
7) The conclusion section is a summary of the important and meaningful results of the article. Some of the conclusions in this paper are inappropriate, and the conclusions are not concise enough.
Minor editing of English language required.
Author Response
On behalf of all authors I would like to thank you for taking your time to read our manuscript and put your comments which allowed to improve the quality of our work. Below you can find our answers related to each of your comments.
- It would be nice if the authors could do more work like EBSD, Texture, TEM, and mechanical testing in this work.
Ad.1. We agree with the reviewer at 100%, such analysis with the use of EBSD, Texture, TEM would significantly improve the scientific point of view in our article, Unfortunately we do not have a proper devices to conduct such analysis. In the case of mechanical testing we did not included further analysis in the case of static (tensile, compressive, bending) properties, fatigue and dynamic testing because it would require additional three combinations of orientation (to keep the reliability of the research). Such an approach would make this paper much bigger. What is more, we wanted to focus mostly on structural phenomena.
- The introduction of this paper is too short. It is also mandatory to add the following references to the text, especially for additive manufacturing of different categories of steel.
- a) Fatigue strength of additively manufactured 316L austenitic stainless steel.
- b) Tensile and fatigue properties of the binder jet printed and hot isostatically pressed 316L austenitic stainless steel.
- c) High temperature study of the evolution of the tribolayer in additively manufactured AISI 316L steel.
- d) Tensile and impact behaviour of thermo mechanically treated and micro-alloyed medium carbon steel bar.
- e) Origin of dislocation structures in an additively manufactured austenitic stainless steel 316L.
Ad.2. The introduction part has been improved based on this comment, however not all points were possible to include due to lack of the mentioned topics in the results according 316L steel obtained via FDM/FFF method.
- “Due to the limited availability of studies describing the 3D printing process of metals, including 316L steel, using the FDM/FFF technology, further research, process refinement, and improvements in the material's performance properties (were carried out ??)”
Remove the word (were carried out ??).
Ad.3. We have removed the mentioned word.
- The full name of CM is missing in the text.
Ad.4. We have added the full name of CM.
- The authors have not discussed why the FDM sample exhibits a higher porosity fraction.
Ad.5. We supplemented the article with this discussion.
- Why is the hardness of FDM/FFF lower than SLM samples?
Ad.6. In the article we included this issue that no additional post-processing treatments were performed on the SLM samples made from 316L steel, which explains their high hardness values. The material microstructure was not unified by any heat treament – so it is on as-built condition (constituted on molten pools)
- The conclusion section is a summary of the important and meaningful results of the article. Some of the conclusions in this paper are inappropriate, and the conclusions are not concise enough.
Ad.7. Based on your and other reviewers comments we rewritten the last part (conclusions).
Round 2
Reviewer 1 Report
The paper has been sufficiently improved to be suitable for publication
Reviewer 4 Report
The paper can be accepted now.